# Trends in smoking initiation and cessation over a century in two Australian cohorts

Alan L. James[1,2☯], Gulser Caliskan[3☯], Giancarlo Pesce[4], Simone Accordini[3], Michael J. Abramson[5], Dinh Bui[6], Arthur W. Musk[7,8†], Matthew W. Knuiman[8], Jennifer L. Perret[9,10], Deborah Jarvis[11], Cosetta Minelli[11], Lucia Calciano[3], Jennie Hui[12,13], Michael Hunter[8,13], Paul S. Thomas[14], E. Haydn Walters[15], Judith Garcia-Aymerich[16,17,18], Shyamali C. Dharmage[9‡], Alessandro Marcon[3‡*]

1 Department of Pulmonary Physiology and Sleep Medicine, Sir Charles Gairdner Hospital, Perth, Western Australia, Australia, 2 Medical School, The University of Western Australia, Perth, Western Australia, Australia, 3 Unit of Epidemiology and Medical Statistics, Department of Diagnostics and Public Health, University of Verona, Verona, Italy, 4 Center for Research in Epidemiology and Population Health, INSERM, Paris-Saclay University, Paris-South University, UVSQ, Villejuif, France, 5 School of Public Health & Preventive Medicine, Monash University, Melbourne, Australia, 6 Allergy and Lung Health Unit, School of Population and Global Health, The University of Melbourne, Melbourne, Australia, 7 Department of Respiratory Medicine, Sir Charles Gairdner Hospital, Perth, Western Australia, Australia, 8 School of Population and Global Health, The University of Western Australia, Perth, Western Australia, Australia, 9 Allergy and Lung Health Unit, Centre for Epidemiology and Biostatistics, Melbourne School of Population and Global Health, The University of Melbourne, Melbourne, Victoria, Australia, 10 Institute for Breathing and Sleep (IBAS), Heidelberg, Melbourne, Australia, 11 National Heart and Lung Institute, Imperial College London, London, United Kingdom, 12 Department of Diagnostic Genomics, PathWest Laboratory Medicine, Perth, Western Australia, Australia, 13 Busselton Population Medical Research Institute, Busselton, Western Australia, Australia, 14 Prince of Wales' Clinical School, UNSW, and Respiratory Medicine, Prince of Wales' Hospital, Randwick, New South Wales, Australia, 15 School of Medicine, University of Tasmania, Hobart, Tasmania, Australia, 16 ISGlobal, Barcelona, Spain, 17 Universitat Pompeu Fabra (UPF), Barcelona, Spain, 18 CIBER Epidemiología y Salud Pública (CIBERESP), Madrid, Spain

☯ These authors contributed equally to this work.
† Deceased.
‡ SCD and AM also contributed equally to this work.
* alessandro.marcon@univr.it

**Data Availability Statement:** All relevant data are within the paper and its Supporting Information files.

## Abstract

### Background

Historical data on smoking can enhance our comprehension of the effectiveness of past tobacco control policies and play a key role in developing targeted public health interventions. This study was undertaken to assess trends in smoking initiation and cessation in Australia for the period 1910–2005.

### Methods

Rates of smoking initiation and cessation were calculated for participants in two population-based cohorts, the Busselton Health Study and the Tasmanian Longitudinal Health Study. The effects of time trends, gender and age group were evaluated.

### Results

Of the 29,971 participants, 56.8% ever smoked. In males, over the period 1910–1999, the rate of smoking initiation in young adolescents remained high with a peak in the 1970s; in

**Funding:** Article processing charges were supported by the special fund at the University of Verona dedicated to Open Access publications. The funder had no role in study design, data collection and analysis, decision to publish, or preparation of the manuscript.

**Competing interests:** The authors have declared that no competing interests exist.

older adolescents it peaked in the 1940s and then declined; in young adults it showed a steady decline. In females, the rate of smoking initiation in young adolescents rose sharply in the 1960s and peaked in the 1970s, in older adolescents it increased throughout the period, and in young adults it declined after 1970. In the period 1930–2005, 27.3% of 9,605 people aged 36–50 years who smoked ceased smoking. Rates of cessation in this age group increased throughout but decreased in males after 1990 and plateaued around 2000 in females.

## Conclusion

Our findings show substantial variation in the efficacy of tobacco control policies across age groups, with a notable lack of success among the younger population.

## Introduction

Cigarette smoking remains a major preventable global health problem [1–3], with increased risks of cancer, chronic airflow obstruction, emphysema, and vascular disease [1, 4]. People who smoke have a reduced life expectancy of approximately 10 years compared with those who do not; cessation of smoking before 40 years of age largely reduces this increased mortality risk [5–7].

Public health campaigns to ban tobacco advertising, increasing the cost of cigarettes, health warnings on packaging and restriction of sales to minors have been successful in reducing the prevalence of smoking [1, 8]. In 2019, the prevalence of current smoking was highest in 40–60 year old Australians, consistent with fewer younger people taking up smoking in recent generations [3].

In Australia, public health groups began raising awareness of the detrimental health effects of smoking after the US Surgeon General's report in 1964, however public health campaigns specifically targeting smoking such as the Quit Campaign only began in 1983, followed by health warnings on packaging and the National Tobacco Campaign in 1997 [9]. These measures have been associated with a steady reduction in the prevalence of smoking. In particular, health warnings, including pictorial representation of smoking related diseases on packaging and laws prohibiting the sale of cigarettes to minors, have all been associated with reduced smoking in teenagers [10]. These approaches and others, including increased cost of cigarettes, mass media controls and introduction of smoke-free environments, have been continued in Australia as part of the National Tobacco Strategies and state-based campaigns.

Despite these gains, the 2022/23 Australian Secondary Students' Alcohol and Drug (ASSAD) survey found that over 10% of school children aged 12–15 in the country reported lifetime smoking [11]. Similarly, up to 15% of younger adolescents (aged 11–15 years) continue to take up cigarette smoking in Europe [2, 12], unlike the falling trends of smoking initiation in older adolescents (16–20 years) and young adults (21–35 years) [12].

Prevalence rates of smoking reflect the balance between smoking initiation and cessation. In Australia, smoking cessation showed a general increase which was predicted to continue [13–15]. In Europe, cessation rates in males have increased over many decades and have also increased in younger females more recently, especially in Northern Europe [16], with similar trends in North America [13, 17]. Knowledge of such trends could help tailor public health interventions by identifying specific groups who will thus experience a disproportionate exposure to tobacco, such as Aboriginal and Torres Strait Islander people, those who are culturally

and racially marginalised, people with mental illness, and people living in rural, regional, and remote areas [3]. Specific groups or conditions where the need for smoking cessation is greatest include pregnancy and breastfeeding, the young and those with pre-existing co-morbidities, particularly coronary heart disease [18].

Understanding historical trends in smoking habits can provide context for current and future public health policies. In this analysis, we made use of two long-standing community-based health studies to assess trends in smoking initiation and cessation over 95 years.

## Methods

### Study design and population

The Tasmanian Longitudinal Health Study (TAHS) investigated all 7 years-old children attending schools in that State in 1968 (the proband cohort) and their siblings [19]. Thirty percent of the cohort now live across all the other states in Australia. Data were obtained on 5,729 probands (participation rate 78.4%) and 12,104 siblings (participation rate 71.6%) who took part in follow-up studies in 2002 and 2007, respectively.

The Busselton Health Study (BHS) began a series of cross-sectional surveys in 1966 in the regional urban area of Busselton in Western Australia [20]. Children and adults were recruited from the general population through schools and public media campaigns between 1966 and 1983 and by media campaigns (including schools) and direct mailing to all local-government area residents on the electoral rolls (voting is compulsory in Australia) from 1990 to 2020 [21]. For the present analysis, data were obtained from eight time points between 1966 and 2015. The number of participants ranged from 1057 (1987) to 5080 (2010–2015) (**S1 Table**); participation rates were fairly high, ranging from 54% (1987) to 91% (1966).

We analysed data from TAHS and BHS separately and after pooling. We performed data pooling for study waves where questionnaire items on smoking habits were comparable. Pooling involved generating aggregated datasets comprising the numbers of cases of smoking initiation/cessation and person-years at risk, categorised by cohort, gender, age of participants (rounded to the nearest integer), and period (calendar year). The pooled datasets are available as **S1 and S2 Datasets**. Comparability of the crude rates of smoking initiation/cessation was also assessed to ensure that data pooling was appropriate.

### Ethics approval and consent to participate

Ethical approval was obtained for this retrospective analysis of pooled data originally collected from the TAHS and BHS studies. The use of TAHS data was approved by the Melbourne School of Population Health Human Ethics Advisory Group (protocol number: 1545792.1). The use of BHS data was approved by the Human Ethics Committee of the University of Western Australia (protocol number: RA/4/1/8288). Consent to participate was not required as all data had been fully anonymized before the statistical analysis was conducted. The authors had no access to information that could identify individual participants. Data were accessed between 1 October 2017 and 30 April 2021.

### Data on smoking

For TAHS, smoking status was derived from the question "In your lifetime, have you smoked at least 100 cigarettes or equal amounts of cigars, pipes or any other tobacco product?". Former smoking was defined, among those who smoked, as having not smoked at all within the last 4 weeks. Age at initiation/cessation was based on the question, "How old were you when you started/stopped smoking?" (**Table 1**).

**Table 1. Questionnaire items on smoking.**

| Study | Smoking Status | Age at initiation | Age at cessation |
|---|---|---|---|
| BHS 1966 | Non-Smoker / Ex-Smoker / Smoker? | Age starting smoking? | Age ceased smoking? |
| BHS 1969 | Are you a Non-Smoker () | At what age did you start smoking? | At what age did you stop smoking for good? |
| | Ex-Smoker () | | |
| | Smoker () | | |
| BHS 1972 | Which of the following best describes your smoking habits? | If applicable, age started | If applicable, age ceased |
| | Non-Smoker () | | |
| BHS 1975 | Cigarette Smoker () | | |
| | Ex-Smoker () | | |
| | Pipe Smoker () | | |
| | Cigar Smoker () | | |
| BHS 1978 BHS 1981 BHS 1987 | Have you ever smoked at least one cigarette per day for as long as one month? No () Yes () | How old were you when you first began to smoke at least one cigarette per day? | How long ago is it since you last smoked at least one cigarette per day? |
| | Do you now smoke at least one cigarette per day? | | |
| | No () Yes () | | |
| BHS 2010 | Have you ever smoked cigarettes? No () Yes () | At what age did you start smoking? | How old were you when you last stopped smoking? |
| | Do you currently smoke manufactured or hand-rolled cigarettes? | | |
| | No () Yes () | | |
| TAHS PROBANDS | In your lifetime, have you smoked at least 100 cigarettes or equal amounts of cigars, pipes or any other tobacco product? | How old were you when you started smoking? | How old were you when you stopped smoking? |
| | NO/YES | | |
| TAHS SIBLINGS | Do you currently smoke (within the last 4 weeks)? | | |

For BHS, slightly different types of questions were available to define smoking status and age at initiation/cessation at distinct study waves (**Table 1**). For participants on more than one occasion, the first available information was used to define age at initiation, and the last available to define cessation [12, 16].

## Statistical analysis

Analyses were performed using STATA 16 software (Stata Corp. College Station, TX, USA). Separate analyses were conducted for males and females. Participants with missing information on smoking status or age at initiation (n = 530) or cessation (n = 320) were excluded (**Fig 1**).

Rates of smoking initiation (per 1,000/year) were calculated retrospectively from childhood to the most recent assessment, as the ratio between the number of people starting smoking and total time at risk (person-years). Participants were considered at risk from age 11 to age at the last study or age 35, which ever came first, since few people reported taking up smoking outside this range [12].

Rates of smoking cessation (per 1,000/year) were calculated as the ratio between the number of people who quit smoking and total time at risk (person-years), defined as years from initiation to cessation for those who quit or to the last questionnaire available for people who were still actively smoking, after excluding life-long non-smoking individuals [16]. Participants were considered at risk from age 16 to age at the last study or age 65, which ever came first, since data were sparse outside this range.

Crude rates of smoking initiation and cessation were reported by decades over the periods 1910–1999 and 1930–2005 respectively, for TAHS and BHS separately and using the pooled dataset. The analysis of smoking initiation was conducted for three separate age groups,

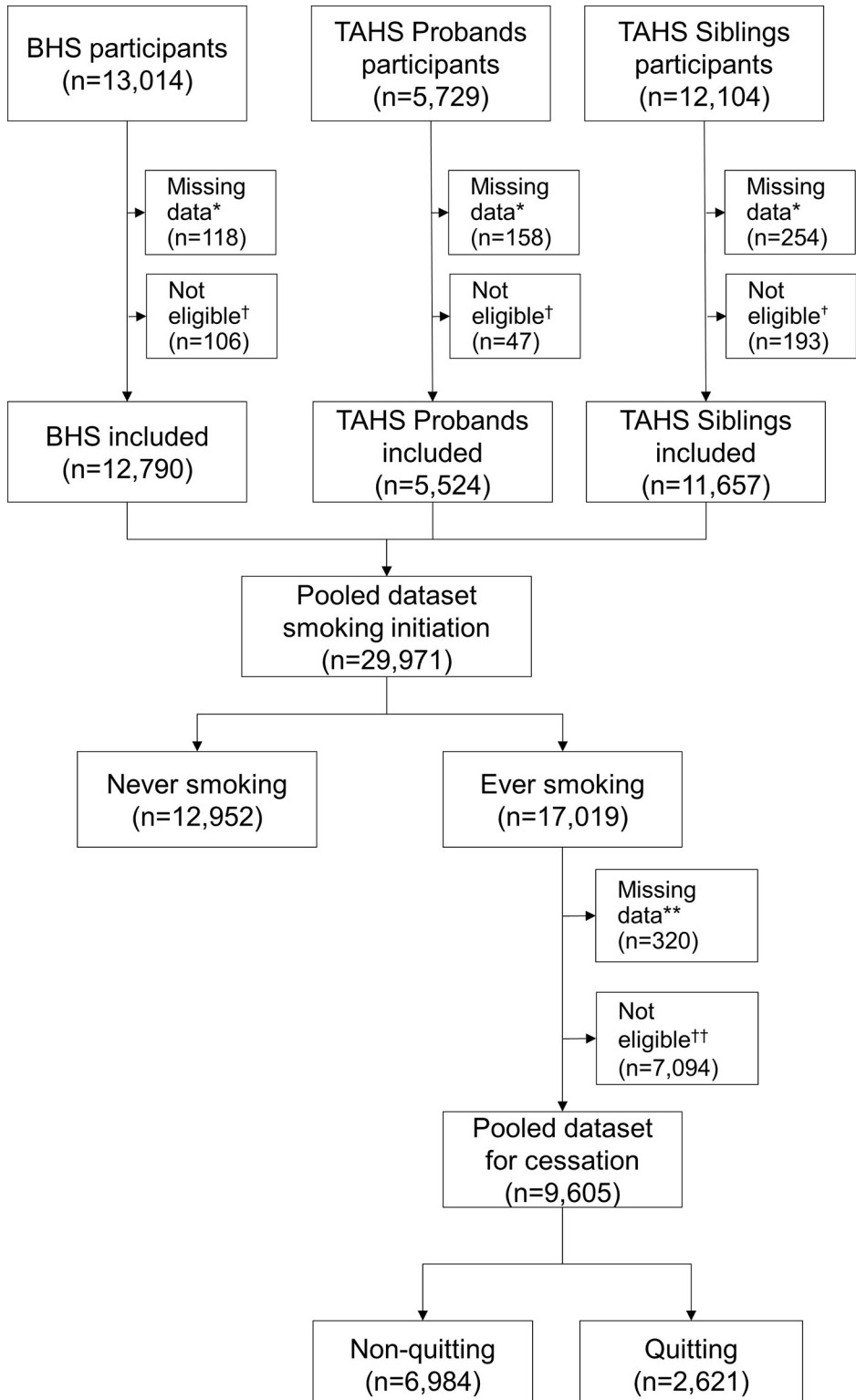

**Fig 1. Study flowchart.** * subjects with missing data on smoking status or age at initiation. † subjects starting smoking outside the eligible age range (11–35 years). ** subjects with missing data on quitting or age at cessation. †† people quitting smoking outside the eligible age range (35–50 years).

referred to as young adolescents (11–15 years), older adolescents (16–20 years), and young adults (21–35 years) [12]. The crude analysis of smoking cessation was conducted for subjects aged 16–35, 36–50, and 51–65 years.

Smoothed trends in smoking initiation/cessation with 95% confidence intervals were estimated using generalized linear models and a negative binomial distribution, a logarithmic link function, and an offset for log person-years. Period (time) was modelled as the main independent variable using natural splines with equally spaced inner knots. The number of knots that provided the best fit according to the Bayesian Information Criterion (BIC) were selected (**S2 Table**). Each analysis was restricted to calendar years with more than 100 person-years at risk to avoid sparse data. As a consequence, the adjusted analysis of smoking cessation was restricted to participants aged 36–50 years (data were sparse in 2000–2005 and 1980–1989 for the 16–35 and 51-65-year age groups, respectively). Multivariable models were adjusted for study group by including a categorical independent variable (coded as TAHS probands, TAHS siblings, and BHS for initiation; for cessation, it was recoded as TAHS and BHS due to sparse data), and age. Age$^2$ was also included to account for non-linearity (except for the analyses on initiation stratified by age group).

As the risk of relapse is high in the first year after quitting, while it drastically drops afterwards [22], we performed a sensitivity analysis defining sustained smoking cessation as having quit smoking for at least two years (with person-years re-calculated accordingly), in order to exclude subjects who might have quit for a shorter time before the survey.

## Results

After exclusions, 29,971 participants made up the pooled dataset for the analysis on smoking initiation (**Fig 1**). **S1 Fig** illustrates the number of BHS and TAHS participants included according to the period of data collection. From BHS, 13,014 participants took part in at least one wave (**Fig 1**) including 2,486 at multiple time points. Overall, there were 15,227 (50.2% female), and the median age at participation was 49 years (**Table 2**).

**S3** and **S4 Tables** compare age at smoking initiation and cessation, respectively, reported at different waves of the BHS study.

There were 17,019 (56.8%) participants who had smoked regularly at some time in their life and mean age at smoking initiation was 17.4 years.

Over the period 1910–1999, the crude rates of smoking initiation were highest at age 15–16 for males (127.1 per 1,000/year) and at age 17–18 for females (98.9 per 1,000/year) (**Table 3**).

In young adolescents, the crude rates of smoking initiation overall were relatively stable in males but were lower in females until 1970 and then increased steeply such that, by 1980–1989, rates of smoking initiation among females had almost reached those in males (87.8 vs 91.2 per 1,000/year) (**S5** and **S6 Tables**). In older adolescents, initiation rates peaked in 1940–49 for males (173.9 per 1,000/year) and 1970–79 for females (107.9 per 1,000/year) and decreased thereafter. In young adults smoking initiation rates steadily decreased for males (**S5 Table**) over the period 1910–1999, from 61.9 to 2.4 per 1,000/year, while smoking initiation rates were consistently lower in females (**S6 Table**), peaking in the 1940s and declining thereafter.

Adjusted smoothed trends of smoking initiation between 1910 and 1980 for all age groups combined (11–35 years) showed decreasing trends for males and increasing trends for females (**Fig 2, panel A**).

Adjusted trends supported the patterns observed for crude rates when stratified by age group (**Fig 3**), with three exceptions.

**Table 2. Number and characteristics of subjects included in the analyses, by study group.**

|  | BHS [a] | TAHS | TAHS | Overall |
|---|---|---|---|---|
|  |  | Probands | Siblings |  |
| Analysis on initiation (1910–1999) |  |  |  |  |
| Subjects (n) | 12,790 | 5,524 | 11,657 | 29,971 |
| Female (%) | 6,817 (53.3) | 2,708 (49.0) | 5,626 (48.3) | 15,151 (50.6) |
| Birth cohort, year (median, min–max) | 1945 (1873–1965) | 1961 (1960–1962) | 1959 (1936–1968) | 1957 (1873–1968) |
| Age, year (median, min–max) | 56 (16–98) | 43 (41–45) | 49 (39–71) | 49 (16–98) |
| Ever smoking (%) | 6,942 (54.3%) | 3,261 (59.0%) | 6,816 (58.5%) | 17,019 (56.8%) |
| Age at initiation, year (mean±SD) | 18.5±5.7 | 16.5±3.7 | 16.6±3.6 | 17.4±4.7 |
| Total years at risk for initiation (age range 11–35 years) | 195,996 | 77,717 | 165,646 | 439,359 |
| Analysis on cessation (1930–2005) |  |  |  |  |
| Subjects (n) | 3,287 | 2,110 | 4,208 | 9,605 |
| Quitting (%) | 1,010 (30.7%) | 457 (21.7%) | 1,154 (27.4%) | 2,621 (27.3%) |
| Age at cessation, year (mean±SD) | 43.1±4.4 | 39.7±2.2 | 42.2±4.0 | 42.2±4.1 |
| Total years at risk for cessation (in age range 36–50 years) | 34,617 | 15,296 | 37,455 | 87,368 |

[a] participants at multiple time points were only considered once; age at last time point reported for descriptive purposes

First, the increasing trend after 1970 in young adolescent males was less evident in the adjusted analysis; for both genders, the adjusted rates also indicated a peak in the 1970s. Second, older female adolescents had a steady increase in smoking initiation. Third, young adult females had a decreasing trend from the 1950s (~30 per 1,000/year) to the 1990s (~0 per 1,000/year).

In the pooled dataset for the analysis of smoking cessation (1930–2005), subjects quitting were 27.3% of people who smoked in the age range 36–50 years (**Fig 1**) and mean age at cessation was 42.2 years (**Table 2**).

The crude rates of smoking cessation were less consistent between TAHS and BHS compared with smoking initiation (**S7 Table**). Nonetheless, for both studies higher crude rates of cessation were seen after 1970. The crude rates of smoking cessation obtained from the pooled analysis are reported in **Table 4**.

The adjusted smoothed trends between 1935 and 2005 were also consistent with this long-term increase in smoking cessation (**Fig 2, panel B**) but highlighted a decrease in cessation among males since 1990, and a plateau for females around 2000.

**Table 3. Crude rates of smoking initiation and person-years at risk for males and females, by age group (1910–1999)[a].**

|  | Males | | Females | |
|---|---|---|---|---|
| Age group (years) | Rate (per 1,000/year) | Person-years | Rate (per 1,000/year) | Person-years |
| 11–12 | 18.7 | 29,446 | 9.7 | 30,109 |
| 13–14 | 53.9 | 28,022 | 33.2 | 29,297 |
| 15–16 | 127.1 | 24,110 | 89.6 | 26,676 |
| 17–18 | 123.9 | 18,215 | 98.9 | 21,899 |
| 19–20 | 61.8 | 14,406 | 46.0 | 18,176 |
| 21–22 | 33.7 | 12,630 | 20.2 | 16,607 |
| 23–26 | 13.3 | 23,355 | 10.1 | 31,359 |
| 27–30 | 6.2 | 22,114 | 6.1 | 29,895 |
| 31–35 | 1.7 | 26,682 | 2.3 | 35,868 |

[a] calculated using the pooled dataset (TAHS + BHS)

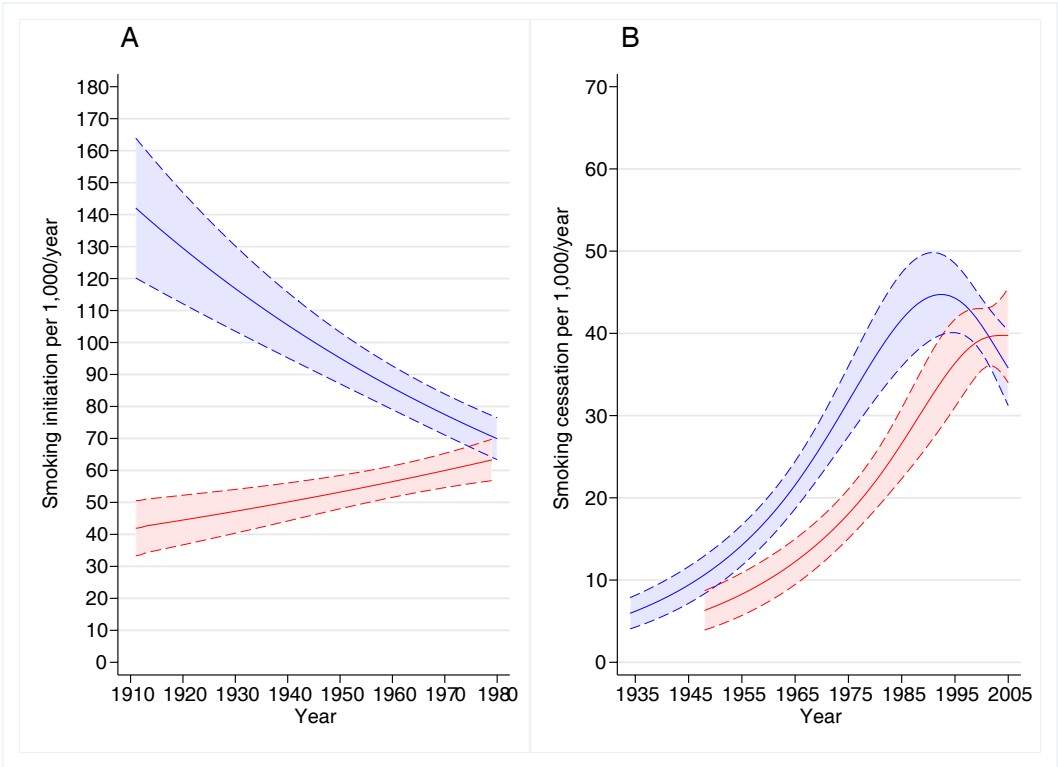

**Fig 2.** Estimated trends in smoking initiation (panel A) and smoking cessation (panel B) with 95% confidence intervals, by sex*. * blue lines: males; red lines: females. Generalized linear models with negative binomial distribution, logarithmic link function, an offset for log person-years, and adjusted for study group, age, and age$^2$; time was modelled using natural splines with equally spaced inner knots (see **S2 Table** for the number of knots).

The sensitivity analysis based on ≥2 years of sustained cessation showed consistent time and age trends in smoking cessation compared to the main analysis, although the magnitude of the rates varied slightly (**S1 Appendix**).

## Discussion

Between 1910 and 1999 up to 57% of people in Australia regularly smoked cigarettes at some stage in their lives, usually starting at 15–16 years of age. In males, rates of smoking initiation have continued to decline in older adolescents and young adults but were high throughout the period. In females, rates of smoking initiation have steadily increased in both young and older adolescents and declined in young adults. The peaks and declines in females show a 30-year lag compared with males so that, in young adolescents, rates of smoking initiation were similar in males and females by the 1980s. Between 1930 and 2005 only 27% of those who smoked in the age range 36–50 years stopped smoking, usually around the age of 42 years. Rates of smoking cessation increased since the 1930s in both males and females although slowed after 1990 in males and after the year 2000 in females.

### Smoking initiation

A study of global trends in smoking initiation in 2019 showed that 82.6% of people who smoke began between the ages of 15 and 24 years and that 18.5% began smoking regularly by the age of 15 years; the relevant figures for Australia were 16.9% for males and 17.1% for females [23].

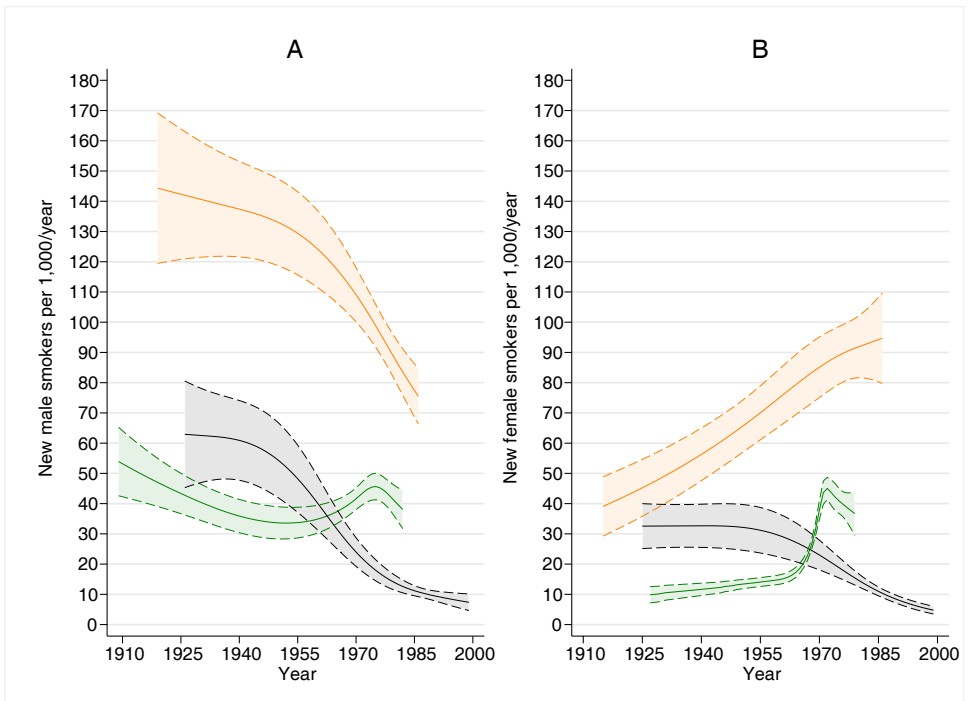

**Fig 3.** Estimated trends in smoking initiation with 95% confidence intervals for males (panel A) and females (panel B), by age group*. * green lines: age 11–15; orange lines: age 16–20; grey lines: age 21–35. Generalized linear models with negative binomial distribution, logarithmic link function, an offset for log person-years, and adjusted for study group, age, and age$^2$; time was modelled using natural splines with equally spaced inner knots (see **S2 Table** for the number of knots).

The Australian 2022–23 National Drug Strategy Household Survey [24] showed that, for those aged 18–24 years, 10.5% of males and 8.3% of females currently smoke, continuing a downward trend from 2001 (34.6% and 29.6% respectively). These data agree with our findings on downward and converging trends for age at initiation of smoking, so that in the most recent birth cohort (1980–89) mean age at smoking initiation was 18.0 years for men and 17.4 for women [25].

A study of smoking initiation in four European regions [12], using a similar analysis to the present study, also showed that between 1970 and 2009, initiation rates were highest in older

**Table 4. Crude rates of smoking cessation and person-years at risk for males and females, by age group (1930–2005)[a].**

|                      | Males                    |              | Females                  |              |
| -------------------- | ------------------------ | ------------ | ------------------------ | ------------ |
| Age group (years)    | Rate (per 1,000/year)    | Person-years | Rate (per 1,000/year)    | Person-years |
| 36–37                | 21.3                     | 10085        | 17.8                     | 8047         |
| 38–39                | 26.6                     | 9522         | 23.3                     | 7698         |
| 40–41                | 41.7                     | 8585         | 42.2                     | 7011         |
| 42–43                | 31.8                     | 7148         | 25.7                     | 5684         |
| 44–45                | 35.5                     | 5071         | 28.9                     | 3828         |
| 46–47                | 24.9                     | 3884         | 24.8                     | 2858         |
| 48–50                | 43.1                     | 4198         | 46.5                     | 2926         |

[a] calculated using the pooled dataset (TAHS + BHS)

adolescents (aged 16–20 years) and decreased in males and to a lesser extent in females, converged in males and females by around 2005, and that initiation occurred at lower ages over time. During periods of observation that overlapped with the present study, similar rates and trends in initiation in males and females were also observed in Europe [12]. These trends continued in general beyond the period of observation of the present study, except that young male and female adolescents showed increasing and higher rates of initiation in the present study than those observed in the European study [12]. The comparable overall trends observed in Europe and Australia align with similar global social and marketing factors, and health policies that influenced smoking initiation. The increased initiation rates in young adolescents in Australia suggests a group that needs long-term monitoring and targeting.

## Smoking cessation

The long-term increase in smoking cessation that was observed from 1935 to 1990 in the present study is promising. However, the slowing down (in males) and plateauing (in females) of cessation rates that was observed by 2005 is concerning.

Using modelling methods and large data bases to estimate current and future prevalence and cessation rates of smoking, Holford et al. showed that cessation rates were higher in later birth cohorts and increased with age in all cohorts in both males and females in the US [13]. However, using similar methods based on Australian data, Gartner et al. found a decrease in cessation rates in younger (20–30 years) males and females in the period 2001–2007 [14]. Trends were variable in the age group 31–50 years during the same period, with an increase in cessation rates in males and a slight decrease in females. In our study, fully adjusted cessation rates for different age groups could not be calculated due to smaller sample sizes. However, the crude rates of cessation showed a decrease in younger (16–35 years) males during 2000–2005 and in older (51–65 years) males and females during 1990–1999, and a continuing upward trend in younger females (16–50 years) (**S7** and **S8** Tables).

Comparing the cessation rates over a similar period (1995–2005) and age range (36–50 years) in our study (**Fig 2**) with those in Europe [16], several trends were evident. Firstly, cessation rates were generally higher in Australia than in Europe, except in North Europe, likely attributable to more widespread or effective implementation of tobacco control policies [16]. Secondly, cessation rates increased in Europe and the US [13, 16] while they changed very little among Australian women and even decreased slightly among Australian men in our study. These discrepancies may be related to our different methodology compared to most reports based on repeated cross-sectional surveys, or to imprecise estimated trends in most recent years due to sparse data. It is also possible that the rates calculated for ages 36–50 years in our study did not apply to a wider age range. Nonetheless, a higher rate of quitting for females compared to males has also been observed in a recent study in Australia [15]. Stable or decreasing cessation rates could be in part related to an increase over time, in our fixed cohort, in the proportion of long-term smoking individuals who face greater challenges when attempting to quit. Thirdly, cessation rates in females were greater than in males in Europe while in Australia they initially showed a lag of about 15 years in females compared with males although were similar by 2005. Data from North America and UK [17] using quit ratios, and Europe [16] showed that the rates of smoking cessation were higher below the age of 40 years in females but higher above the age of 40 in males. Peaks of cessation across all European regions in their mid to late 20s in females with a more blunted peak in males of a similar age [16], likely represented smoking cessation related to pregnancy and child rearing. These trends suggest that focusing public health programs on specific groups is effective, and that targeting young adult males may be a strategy to enhance smoking cessation in Australia.

## Strengths and limitations

The main strength of the present study was the use of two large, representative population studies with repeated measures over up to 70 years and birth cohorts extending over almost a century [19, 26].

However, we made secondary use of data that were not specifically collected for investigating temporal trends in smoking habits. We acknowledge the potential bias arising from imbalance of participants across ages and birth cohorts due to attrition of the cohorts over time related to dropouts and mortality [15]. Our historical analysis regrettably lacks data on the recent epidemic of use of e-cigarettes and other alternative nicotine-containing products.

There were some differences between the BHS and TAHS, including wording of questionnaires, age of participants and periods of data collection. Analyses therefore were undertaken separately for each study group and, in pooled analysis, with adjustments for study group. To avoid sparse data we did not assess the standardised cessation rates by age group, which were likely to differ in both males and females [16, 17]. Given the long periods of observation, there were changes in the population of participants over time including older age of reported smoking cessation, possibly due to smoking persistence or relapses in people with higher levels of nicotine addiction (**S3 Table**). Recall bias may also affect reported ages of smoking initiation and/or cessation. Although the mean age of smoking initiation remained consistent between waves of the BHS study, approximately 30% reported age of initiation greater or lesser by more than 2 years at the second wave (**S2 Table**). Likewise, the mean age at smoking cessation remained similar; however roughly 40% participants exhibited a variation of more than 1 year between the two study waves (**S3 Table**).

BHS and TAHS samples may not be fully representative of the broader Australian population, which could limit generalisability of our results. Regional centres have 3–5% higher prevalence of current smoking compared with major cities [29]. Compared with national Australian data [27, 28], the Busselton population showed higher proportions of adults formerly smoking in both genders between 1973 and 1984, and higher proportions of lifetime non-smoking individuals in women but not men [21]. Compared with a sample from the Australian National Health Survey 2011, Busselton adults aged 46–64 years studied between 2005 and 2007 were similar on a number of key health indicators, however the prevalence of current tobacco smoking in Busselton was 10%, lower than the national average for this age group at the time (15%) [29]. Finally, the trends seen in smoking initiation and cessation may depend to some extent on changing demographics in the Busselton population over the last century.

## Implications of findings

The determinants of smoking initiation in high income countries are social and behavioural and include location, having friends or family who smoke, alcohol use, perceived relief of stress, and the influence of marketing [30–32]. These widespread influences on initiating smoking are reflected in the upward trends in smoking initiation observed in the young adolescents in the present study. Efforts to counteract these influences [1, 8, 33–35] likely account for the downward trends in smoking initiation in young adults and older adolescent males in our study. The trends in smoking initiation and cessation we have observed in two Australian samples tend to match those seen in other affluent countries and regions [12, 16]. This suggests that similar social and public health influences affect these behaviours. It is notable that the high rates of smoking initiation in males, particularly in those between 16–20 years of age, coincide with the first and second World Wars (1910–1919 and 1940–1949 in **S4 Table**). Further peaks in smoking initiation after the 1960s likely reflect evolving social attitudes of the times and the implementation of more aggressive marketing strategies for smoking. The lags

in smoking initiation and cessation rates between females and males observed in the present study may also reflect social pressures. The higher rates of initiation in Australia in young adolescents, compared with Europe, and the slowing of cessation rates observed in the early part of this century suggest that targeted public health measures are needed in these areas.

Although considerable gains have been made in addressing the smoking epidemic, at least in the more affluent countries, studies in Australia, Europe and North America continue to show high rates of current smoking with a gradual reduction in age of smoking initiation and a decline in rates of smoking cessation. With a few notable exceptions, the prevalence of current smoking has decreased globally in 15–24 year olds, predominantly due to a reduction in smoking initiation [23], although smoking cessation may contribute in younger females [16, 17].

The adoption of the WHO Framework Convention on Tobacco Control has been associated with considerable reductions in the prevalence of smoking [36]. However, population growth has resulted in an increase in the total number of people who smoke globally [23]. Persisting high levels of current smoking may be due to low rates of cessation in susceptible individuals with higher levels of nicotine addiction, and/or a reduction in public health efforts and funding [37] to counter smoking initiation, particularly in young adolescents from low socioeconomic groups. It is also recognised that smoking initiation rates are higher and smoking cessation rates are lower in specific groups such as indigenous communities [38] and in those with long-term mental illness [39]. Multiple factors such as high levels of exposure to smoking and marginalization contribute to the challenge of reducing smoking prevalence in Indigenous communities, although substantial progress has been made in the last decades [40]. The development of acceptable and culturally appropriate interventions is required to address these complex, overlapping issues.

Over the course of the last century, factors that have contributed to smoking have included global conflicts, high volume manufacture of cigarettes, brand recognition and social marketing. More recently e-cigarettes and vaping have arisen as an additional possible gateway to cigarette smoking [41, 42]. e-Cigarette use in the past month has increased from 4.2% in 2017 to 15.7% in 2022, particularly in young adults [3].

## Conclusions

The present study has provided a historical view of the trends in smoking initiation and smoking cessation over the last century using standardised rates for comparisons across regions and countries. It shows striking period, age and gender effects, with a lesser effect of cohort, suggesting that the efficacy of tobacco control policies has not been uniform across all age groups, with a notable lack of success among the younger population since the 1970s.

The changes observed over time represent a balance of global social, mental health, political and marketing pressures on the one hand and public health measures on the other. The comparisons between Australia, Europe and other high-income countries demonstrate similar patterns likely related to global influences of social, media and public health campaigns. The persistence of a health inequity due to smoking, particularly among susceptible groups, suggests the need for identifying more specific risk factors for smoking initiation [34], developing better targeted approaches for smoking cessation [43] and addressing the emerging risks associated with e-cigarettes and increased mental health disorders.

## Supporting information

**S1 Fig. Number of BHS and TAHS participants included in the analysis according to the period of data collection.**
(DOC)

**S1 Appendix. Sensitivity analysis on sustained smoking cessation.**
(DOC)

**S1 Dataset. Minimal data set to replicate the analyses on smoking initiation[a].** [a] variables included: cohort, sex, A (age), P (calendar year), D and pop (cases of smoking initiation and person-years at risk).
(CSV)

**S2 Dataset. Minimal data set to replicate the analyses on smoking cessation[a].** [a] variables included: cohort, sex, A (age), P (calendar year), D and pop (cases of smoking cessation and person-years at risk), D2 and pop2 (sensitivity analysis: cases of sustained smoking cessation and person-years at risk).
(CSV)

**S1 Table. Distribution of the characteristics of BHS participants by study wave.**
(DOC)

**S2 Table. Values of the Bayesian Information Criterion (BIC) for the different numbers of knots tested in the natural spline functions for period, by analysis and sex.**
(DOC)

**S3 Table. Comparison of age at smoking initiation reported at different BHS waves[a].** [a] analysis restricted to the subjects who reported to be ever smokers at both waves under comparison.
(DOC)

**S4 Table. Comparison of age at smoking cessation reported at different BHS waves[a].** [a] analysis restricted to the subjects who reported to be quitters at both waves under comparison.
(DOC)

**S5 Table. Crude rates of smoking initiation per 1000/year and person-years at risk for males by age group, cohort and period[a].** [a] cells with less than 100 person-years at risk were omitted.
(DOC)

**S6 Table. Crude rates of smoking initiation per 1000/year (and person-years at risk) for females by age group, cohort and period[a].** [a] cells with less than 100 person-years at risk were omitted.
(DOC)

**S7 Table. Crude rates of smoking cessation per 1000/year (and person-years at risk) for males, by cohort and period.**
(DOC)

**S8 Table. Crude rates of smoking cessation per 1000/year (and person-years at risk) for females, by cohort and period.**
(DOC)

## Author Contributions

**Conceptualization:** Alan L. James, Simone Accordini, Shyamali C. Dharmage, Alessandro Marcon.

**Data curation:** Gulser Caliskan, Giancarlo Pesce.

**Formal analysis:** Gulser Caliskan, Alessandro Marcon.

**Investigation:** Alan L. James, Gulser Caliskan, Giancarlo Pesce, Simone Accordini, Shyamali C. Dharmage, Alessandro Marcon.

**Methodology:** Gulser Caliskan, Giancarlo Pesce, Simone Accordini, Lucia Calciano, Alessandro Marcon.

**Resources:** Alan L. James, Simone Accordini, Michael J. Abramson, Arthur W. Musk, Matthew W. Knuiman, E. Haydn Walters, Shyamali C. Dharmage.

**Supervision:** Alan L. James, Alessandro Marcon.

**Writing – original draft:** Alan L. James, Gulser Caliskan, Jennie Hui, Shyamali C. Dharmage, Alessandro Marcon.

**Writing – review & editing:** Alan L. James, Gulser Caliskan, Giancarlo Pesce, Simone Accordini, Michael J. Abramson, Dinh Bui, Arthur W. Musk, Matthew W. Knuiman, Jennifer L. Perret, Deborah Jarvis, Cosetta Minelli, Lucia Calciano, Jennie Hui, Michael Hunter, Paul S. Thomas, E. Haydn Walters, Judith Garcia-Aymerich, Shyamali C. Dharmage, Alessandro Marcon.

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
