## [Decision Letter · Decision Letter 0]

21 Feb 2024

PONE-D-24-00230Trends in smoking initiation and cessation over a century in two Australian cohortsPLOS ONE

Dear Dr. Marcon,

Thank you for submitting your manuscript to PLOS ONE. After careful consideration, we feel that it has merit but does not fully meet PLOS ONE’s publication criteria as it currently stands. Therefore, we invite you to submit a revised version of the manuscript that addresses the points raised during the review process. In particular, you should address the methodological comments by our reviewers, the question of representativeness and generalizability of results, potential biases associated with restriction to ages 35 - 50 years, and the possible effects and timing of tobacco-control measures in Australia. ** **

We look forward to receiving your revised manuscript.

Kind regards,

Thomas Behrens

Academic Editor

PLOS ONE

“Article processing charges were supported by the special fund at the University of Verona dedicated to Open Access publications”

“Article processing charges were supported by the special fund at the University of Verona dedicated to Open Access publications.”

“Article processing charges were supported by the special fund at the University of Verona dedicated to Open Access publications”

Reviewers' comments:

Reviewer's Responses to Questions

**Comments to the Author**

1. Is the manuscript technically sound, and do the data support the conclusions?

Reviewer #1: Yes

Reviewer #2: Partly

Reviewer #3: Partly

2. Has the statistical analysis been performed appropriately and rigorously? 

Reviewer #1: Yes

Reviewer #2: Yes

Reviewer #3: No

3. Have the authors made all data underlying the findings in their manuscript fully available?

Reviewer #1: No

Reviewer #2: Yes

Reviewer #3: Yes

4. Is the manuscript presented in an intelligible fashion and written in standard English?

Reviewer #1: Yes

Reviewer #2: Yes

Reviewer #3: Yes

5. Review Comments to the Author

Reviewer #1: The authors describe smoking initiation and cessation among two Australian cohorts (Tasmanian Longitudinal Health Study (TAHS) and Busselton Health Study (BHS)) who were 7 years old in 1966 and 1968 and followed for 40-50 years.

The most recent data were collected in 2005, nearly 20 years ago and well before the e-cigarette era, which radically changed youth nicotine addiction patterns. The authors briefly note this in the abstract and in the main text at line 391. The development of e-cigarettes, however, needs to be more carefully taken into account when interpreting the relevance of the results to public health policy making today. This does not invalidate the findings as far as they go, but require more cautious conclusions about current relevance of those findings.

The results were pooled into a single analysis. There is nothing wrong in principle with doing this, but the smoking questions were slightly different in the two studies. How they were combined and any associated limitations and uncertainties need to be described in more detail.

The authors censored data for people who stopped smoking before age 35. This is a major problem, since many adolescents who experiment with cigarettes stop by young adulthood. To provide an accurate balance between initiation and cessation, the full range of cessation ages need to be included.

How were the statistical adjustments made (line 186)?

RE: Data availability: The authors state, "All relevant data are within the manuscript and its Supporting Information files," but I don't see the actual raw data. The raw data also needs to be made available per PLOS ONE policies.

Reviewer #2: Trends in smoking initiation and cessation over a century in two Australian cohorts (PONE-D-24-00230)

Thank you for the possibility to peer-review the abovementioned study. The aim of the study was to investigate the trends of smoking initiation and cessation in Australia between 1910 and 2005. The topic is important as smoking remains one of the largest threats for public health. In this study, time trends, as well as gender and age effects were evaluated within nearly 30.000 participants of two population-based cohorts. The findings highlight that tobacco control policies have had less success among younger populations.

I would like to point out a few questions and comments:

1. The authors mention in the introduction that, according to previous literature, public health campaigns to ban tobacco advertising, increasing the cost of cigarettes, health warnings on packaging and restriction of sales to minors have been successful in reducing the prevalence of smoking. Further, they conclude that the efficacy of tobacco control policies has not been uniform across all age groups, with a notable lack of success among the younger population since the 1970s. I was wondering what tobacco control policies have been made in Australia during the studied time period? Can you see the effects of certain policy changes? Have some policy changes been more effective than others? Can you say if some policy changes have better effects on the younger / older population than others?

2. The study was conducted in Australia. How do the authors consider the generalizability of the findings into other countries?

3. What are the current tobacco policies in Australia? Can the authors make more explicit policy recommendations based on their findings?

4. Could the authors please add the (original) number of participants in the BHS cohort to the “study desing and population” paragraph?

5. The eligible age range for smoking cessation was 35-50 years to restrict to participants with a similar age distribution in the two cohorts. The age range for smoking initiation (11-35 years) was based on previous literature. Could the authors give information from previous literature, how many smokers do stop smoking at this chosen age? Does this age range cause selection bias to the results? How many smokers give up smoking, e.g., already in their late twenties or early thirties?

6. The authors state as the main finding of the study: “Between 1910 and 1999 up to 57% of people in Australia regularly smoked cigarettes.” How representative are these two cohorts of the total Australian population? Can the authors tell a bit more about the selection criteria for the Busselton Health Study cohort? Are children who attended school in Tasmania in 1968 similar to the total Australian population? There was some loss to follow up, does this affect the representativeness?

7. The authors recommend targeting young adult males to enhance smoking cessation in Australia. However, as discussed above, the study only investigated smoking cessation after the age of 35 years.

Reviewer #3: # Review summary

Thank you for the opportunity to review the manuscript 'Trends in smoking

initiation and cessation over a century in two Australian cohorts'.

The manuscript reports on the rates of initiation and cessation estimated from

longitudinal cohort studies conducted in Australia beginning in the latter part

of the 20th century. The authors estimated trends in the rate of initation and

cessation of regular smoking amongst those who never (regularly) smoked and

those who do smoke, respectively.

The statistical analysis is not quite complete, with some additional work

needed to satisfy a reasonable scientific standard for this type of analysis. I

have a few criticisms/suggestions for the introduction and discussion as the

literature review does not accurately reflect the available knowledge for the

Australian context, and the strengths and limitations of the study could be

amended. I'll provide feedback section-by-section. I believe the extent of

the amendments would be best described as 'major revisions'.

## Introduction

1st paragraph: The following reference is relevant to the 2nd sentences,

providing an Australian estimate on the number of years of life lost for those

who continue to smoke:

- Banks, E., Joshy, G., Weber, M.F. et al. Tobacco smoking and all-cause

mortality in a large Australian cohort study: findings from a mature

epidemic with current low smoking prevalence. BMC Med 13, 38 (2015).

https://doi.org/10.1186/s12916-015-0281-z

In addition; it was also estimated (contemporaneously to reference [5] in the

manuscript) that those in the UK who quit before age 40 largely avoid the

loss:

- Pirie, K., Peto, R., Reeves, G. K., Green, J., & Beral, V. (2013). The 21st

century hazards of smoking and benefits of stopping: a prospective study of

one million women in the UK. The Lancet, 381(9861), 133-141.

https://doi.org/10.1016/S0140-6736(12)61720-6

2nd paragraph: References [2,7] are specific to the European setting, however

this is not clear in the text. In fact, there are ongoing surveys which have

estimates for adolescent smoking in Australia, the 'ASSAD' survey, see for

example this report (either the 2017 or 2023 release should have been

referenced):

https://www.health.gov.au/resources/publications/secondary-school-students-use-of-tobacco-and-e-cigarettes-2022-2023?language=en

3rd paragraph: Another reference that is important to the (global) understanding

of initiation and cessation rates is the study of Holford et al, in 2014:

- Holford, T. R., Levy, D. T., McKay, L. A., Clarke, L., Racine, B., Meza, R.,

... & Feuer, E. J. (2014). Patterns of birth cohort–specific smoking

histories, 1965-2009. American journal of preventive medicine, 46(2),

e31-e37.

which has also been picked up in Canada as a basis for their own model (but not

of particular interest for this paper).

The authors highlight certain priority populations, however these have been

revised as of 2021 in Australia (nor does reference [3] provided by the

authors mention those in jail). Care should be taken that Aboriginal and Torres

Strait Islander people is capitalised correctly - the Australian co-authors on

this paper should be able to provide guidance on these sensitive practises.

The National Preventive Health Strategy 2021-2030 for Australia identifies the

following priority populations:

- Aboriginal and Torres Strait Islander people

- Culturally and linguistically diverse (CALD)

- LGBTQI+

- people with mental illness

- people of low socioeconomic status

- people with disability

- rural, regional, and remote

Source:

https://www.health.gov.au/resources/publications/national-preventive-health-strategy-2021-2030?language=en

I would also advise against the wording 'resistant to cessation'; this is

suggestive of a white, straight, able-bodied 'default' that is superior in some

way to the 'other'.

Instead I would rephrase the sentence on lines 94-97 to "Knowledge of such trends

could help tailor public health interventions by identifying specific groups who

will thus experience a disproportionate exposure to tobacco; such as Aboriginal

and Torres Strait Islander people, culturally and racially marginalised, people

with mental illness, and people living in rural, regional, and remote areas."

The ethics of _only_ discussing disparity and disadvantage is also

questionable, i.e. progress should also be included:

- Lovett R, Thurber KA, Wright A, Maddox R, Banks E. Deadly progress: changes

in Australian Aboriginal and Torres Strait Islander adult daily smoking,

2004-2015. Public Health Res Pract. 2017;27(5):e2751742.

https://doi.org/10.17061/phrp2751742

4th paragraph: The Australia context is not as scarcely studied as the authors

suggest. In addition to the references above (the study on mortality by Banks,

the ASSAD survey), there is also a study on cessation rates by Gartner et al,

from 2009, along with a newer pre-print that extends the work to cover the

better part of 80 years:

- Gartner, C. E., Barendregt, J. J., & Hall, W. D. (2009). Predicting the

future prevalence of cigarette smoking in Australia: how low can we go and

by when?. Tobacco control, 18(3), 183-189.

- Wade, S., Weber, M. F., Sarich, P., Vaneckova, P., Behar-Harpaz, S., Ngo,

P. J., ... & Caruana, M. (2022). Bayesian calibration of simulation models:

A tutorial and an Australian smoking behaviour model. arXiv preprint

arXiv:2202.02923.

And there is the following study on long-term trends in cross-sectional surveys

of Australians (incl. the age at initiation, and the overall number that

initiate):

- Vaneckova P, Wade S, Weber M, Murray JM, Grogan P, et al. (2021)

Birth-cohort estimates of smoking initiation and prevalence in 20th century

Australia: Synthesis of data from 33 surveys and 385,810 participants. PLOS

ONE 16(5): e0250824. https://doi.org/10.1371/journal.pone.0250824

## Methods

### Data on smoking

Relapse is common after only 1 month of cessation, see:

Herd, N., Borland, R., & Hyland, A. (2009). Predictors of smoking relapse by

duration of abstinence: findings from the International Tobacco Control (ITC)

Four Country Survey. Addiction, 104(12), 2088-2099.

therefore there is likely a lot of misclassification amongst those who had

quit 1 or 2 years before the most recent measurement. If possible, the

observation of smoking status could be censored to 2 years prior to the

most recent survey in either the main analysis, or as a sensitivity analysis.

### Statistical analysis

The number of knots considered and the BIC for each model should be reported.

The predictions of the model should also be compared to the data used; e.g.

for each data point, a _prediction_ interval (not confidence interval) should be

provided, then outliers and the coverage of the intervals should be assessed.

## Discussion

1st sentence: Revise, the data suggests that 57% of people regularly smoked

over the entire period, but at any one point in the period it may have been

lower or higher than this.

2nd sentence: Revise: "Between 1930 and 2005 only 27% of those who smoked in

the age range 35-50 years stopped smoking"

### Smoking initiation

1st parapgraph: Reference [13] in the manuscript did not use any local sources

of data on age at initiation (see its supplementary material); it would be

better to use the published sources of information which include the National

Drug Strategy Household Survey (NDSHS) 2019 (reference [3] in manuscript) or

tThe reference to Vaneckova et al, 2019, I provided earlier also provides local

data on the proportion of those who (ever) smoked that start smoking by a given

age. The NDSHS 2019 also indicates that fewer than 20% of 18-24 year olds smoke

(10% daily, and 7% occasional), therefore the final sentence should be amended.

## Smoking cessation

Line 316: remove repeated 'effective'.

Line 321: Reference [9] does not estimate quit rates; therefore the comparison

is not valid.

A comparison here could be made to the results provided in the pre-print above

(Wade et al, 2022). Perhaps, more importantly, to other studies such as Holford

et al, 2014, or Gartner et al, 2009.

A critical missing piece of context is that the National Tobacco Campaign

(1997-2001) was in full effect during the period where it has been estimated in

this manuscript that cessation rates were in decline. This requires some

explanation; either the authors would need to substantiate why cessation rates

were in decline or explain why their model disagrees with a fairly reasonable

expectation that they would (likely) be increasing.

One potential explanation I can offer is the reliance on adjustment for age

(and age^2) in a period where less data on cessation from younger smokers is

available (i.e. fewer eventual relapsers are reporting that they had quit). As

mentioned above, a sensitivity analysis which censors observations of cessatio

n events 2 years before the surveys might be helpful here.

### Strengths and limitations

The age-stratified analysis of initiation doesn't appear to account for the

differences in distribution of age in later periods (e.g. for the period

1990-1999, no survey contained an individual aged 21 in that period according

to Table 2); thus the analysis would under-estimate the initiation rate.

Another (likely) source of bias in the initiation rate is survivor bias. This

would lead to under-estimation of the initiation rate in older cohorts.

Misclassification of those who ever smoked (versus never smoked) would also

lead to under/mis-estimation of initiation rates in earlier cohorts; this

misclassification has been observed in the Busselton study itself:

- Visalpattanasin, P., Wearne, K. L., & Armstrong, B. K. (1987). Trends in the

uptake of smoking in Busselton, Western Australia. Australian and New

Zealand Journal of Public Health, 11, 21-s.

Wade et al, 2022, shows (Table S2-10,S2-12) that not accounting for this effect

can materially impact the estimate of both the proportion that initiate within a

cohort and the cessation rate in a calendar year.

An obvious, but not stated, limitation is that the cohorts are not random

samples from the complete Australian population; they are more representative

of 'regional' Australia than metropolitan Australia and given that regional

status is an influence on smoking behaviour - this suggests some care should be

taken when generalising to the Australian population.

### Implications

Line 382: The 'hard core' smokers hypothesis is not considered infallible, see

- Skinner, A., Occhipinti, JA. & Osgood, N.D. A dynamic modelling analysis of

the impact of tobacco control programs on population-level nicotine

dependence. Sci Rep 11, 1866 (2021).

https://doi.org/10.1038/s41598-021-81460-9

Line 385-387: At face value I agree that initiation and cessation rates are

less favourable in Aboriginal and Torres Strait Islander communities, this was

not the focus of reference [23] nor is it covered in [24] for mental illness.

Enduring high prevalence in older populations, described in Lovett et al, 2017,

are supportive of the need for more focus on cessation in Aboriginal and

Torres Strait Islander people. I would rework this last sentence using the

following text from the Conclusions of [23] as a starting point, with some

simplifications needed to put it in the context of the study in the manuscript:

> The challenges for indigenous people are much greater and include poverty,

marginalization, challenges in accessing resources, high rates of smoking, and

acceptance of smoking in families and communities. This review has underscored

the complexity of achieving smoking cessation and the need to collaboratively

develop interventions that are acceptable and appropriate to local

populations.

Line 393: A more up-to-date reference on the use of vaping amongst youth in

Australia is provide by the ASSAD survey I mentioned above, e.g. 15.7% of

students reported using vapes in the past month in 2022, up from 4.2% in 2017.

## Conclusions

Line 411: I'd leave out 'political conflict' unless further commentary on the

difficulty in adoption of the FCTC, re-commitment to MPOWER, and recent changes

in vaping policy are provided in the Discussion.

I'd also pivot away from saying 'unacceptably high level of tobacco use' when

this is being written by and from the perspective of people who do not belong

to the groups being talked about (it could read as an admonishment rather than

coming from a place of empathy). An alternative might be "The persistence of

(health) inequity caused by smoking amongst marginalised people suggests..."

although I recognise it isn't the plainest English.

# Summary of review

The key changes to the analysis I would recommend are:

1. A sensitivity analysis that considers censoring quit events 2 years earlier,

or use this as a main analysis (and include the uncensored as a sensitivity

analysis).

2. Report the number of knots considered and the BIC of each model tested.

3. Provide prediction intervals and overlay the observed data for the main

analysis model.

Depending on these results, I expect that some of the conclusions may need

moderation.

The other issues raised in each section can be addressed via amendments to the

text.

Lastly, I would amend some of the language used throughout, replacing 'smoker'

with 'person who smoked' or similar, as per guidance I have received about

labelling and stigmatisation, also see:

- Hefler M, Durkin SJ, Cohen JE, et alNew policy of people-first language to

replace ‘smoker’, ‘vaper’ ‘tobacco user’ and other behaviour-based labels.

Tobacco Control 2023; 32:133-134.

- Renee D Goodwin, Lisa K Walker, Time to Stop Using the Word “Smoker”:

Reflecting on the Role of Language in Advancing the Field of Nicotine and

Tobacco Research, Nicotine & Tobacco Research, Volume 24, Issue 12,

December 2022, Pages 1847–1848.

https://doi.org/10.1093/ntr/ntac218

6. PLOS authors have the option to publish the peer review history of their article (what does this mean?). If published, this will include your full peer review and any attached files.

Reviewer #1: No

Reviewer #2: No

Reviewer #3: **Yes: **Stephen Wade

---

## [Author Response · Author response to Decision Letter 0]

3 May 2024

A Point-by-point reply letter is attached

---

## [Decision Letter · Decision Letter 1]

12 Jun 2024

PONE-D-24-00230R1Trends in smoking initiation and cessation over a century in two Australian cohortsPLOS ONE

Dear Dr. Marcon,

Thank you for submitting your manuscript to PLOS ONE. After careful consideration, we feel that it has merit but does not fully meet PLOS ONE’s publication criteria as it currently stands. Therefore, we invite you to submit a revised version of the manuscript that addresses the points raised during the review process.

We look forward to receiving your revised manuscript.

Kind regards,

Billy Morara Tsima, MD MSc

Academic Editor

PLOS ONE

Journal Requirements:

Reviewers' comments:

Reviewer's Responses to Questions

**Comments to the Author**

1. If the authors have adequately addressed your comments raised in a previous round of review and you feel that this manuscript is now acceptable for publication, you may indicate that here to bypass the “Comments to the Author” section, enter your conflict of interest statement in the “Confidential to Editor” section, and submit your "Accept" recommendation.

Reviewer #1: (No Response)

Reviewer #2: All comments have been addressed

Reviewer #3: All comments have been addressed

2. Is the manuscript technically sound, and do the data support the conclusions?

Reviewer #1: Partly

Reviewer #2: Yes

Reviewer #3: Yes

3. Has the statistical analysis been performed appropriately and rigorously? 

Reviewer #1: I Don't Know

Reviewer #2: Yes

Reviewer #3: Yes

4. Have the authors made all data underlying the findings in their manuscript fully available?

Reviewer #1: Yes

Reviewer #2: No

Reviewer #3: Yes

5. Is the manuscript presented in an intelligible fashion and written in standard English?

Reviewer #1: Yes

Reviewer #2: Yes

Reviewer #3: Yes

6. Review Comments to the Author

Reviewer #1: Line 92: The Surgeon General Report was published in 1964.

I still worry about drawing strong conclusions about prevention programs today based on such old data.

Line 543-544: Drop this statement. It isn’t really based on your data and doesn’t add anything.

I still don’t understand how the applicants actually did the pooling. It sounds like they just dumped all the data into a single file. Please be more precise.

Also, did the surveys account for complex survey design? If so, how was that considered in the analysis and pooling?

Reviewer #2: (No Response)

Reviewer #3: The authors have comprehensively and accurately responded to all my feedback, with only one minor exception that I'll put on the record to help the authors in any further investigation. Otherwise, the manuscript is worthy of publication and the discussion provides interesting comparison of trends in starting and stopping smoking between settings (US, Europe, Australia).

# Minor feedback

Original feedback: C3.8) The predictions of the model should also be compared to the data used; e.g. for each data point, a _prediction_ interval (not confidence interval) should be provided, then outliers and the coverage of the intervals should be assessed.

Author's response: We have conducted a descriptive analysis on temporal trends in the rates of smoking initiation and smoking cessation using natural splines to model time. The analysis was aimed at describing and visualizing the observed temporal patterns in the study area. For such purposes, we believe it was correct to report confidence intervals, which quantify the uncertainty around the estimated rates based on sample data, under the assumption that the underlying model was correct. Since we were not making out-of-sample predictions or forecasting future observations, we believe we do not need to estimate a prediction interval, which would be used to quantify the uncertainty around a future observation or out-of-sample prediction by accounting for both the uncertainty in the parameter estimate and the inherent variability of future observations. Forecasting was beyond the scope of our analysis, considering the study design and sparse data issues for more recent periods.

Reply: I would still recommend (perhaps in future investigations) that within-sample predictive checks are done on models. This is not to assess how the model performs out of sample. For example, if less (or more) than 95% of the (within sample) prediction intervals contain the observed value, then it is plausible/likely that the model is missing detail or (in very rare cases) was not fit correctly - this also has bearing on whether the confidence intervals themselves have the stated coverage.

Even though your analysis is not Bayesian, a relevant resource is Bayesian Data Analysis by Gelman:

Gelman, A., Carlin, J. B., Stern, H. S., Dunson, D. B., Vehtari, A., & Rubin, D. B. (2014). Bayesian Data Analysis.

# Data availability

I'd recommend confirming that the cell-sizes in the spreadsheet provided do not violate any data confidentiality or other Ethics agreements about using the data. Line data does not need to be made available according to my own understanding of the PLOS ONE data policy.

# Some fixes for references

1. I spotted some inconsistency with shortened journal name vs full name (and capitalisation of PLoS ONE vs PLOS ONE)

2. Check capitalisation in reference 26, and title case (?) in 28 ,34, 41, and 43.

3. pdf links seemed not to work (probably just a rendering issue).

7. PLOS authors have the option to publish the peer review history of their article (what does this mean?). If published, this will include your full peer review and any attached files.

Reviewer #1: No

Reviewer #2: No

Reviewer #3: **Yes: **Stephen Wade

---

## [Author Response · Author response to Decision Letter 1]

1 Jul 2024

Please see the responses to reviewers letter attached

---

## [Editor Report · Decision Letter 2]

4 Jul 2024

Trends in smoking initiation and cessation over a century in two Australian cohorts

PONE-D-24-00230R2

Dear Dr. Marcon,

We’re pleased to inform you that your manuscript has been judged scientifically suitable for publication and will be formally accepted for publication once it meets all outstanding technical requirements.

Kind regards,

Billy Morara Tsima, MD MSc

Academic Editor

PLOS ONE